# Metabolic Responses, Cell Recoverability, and Protein Signatures of Three Extremophiles: Sustained Life During Long-Term Subzero Incubations

**DOI:** 10.3390/microorganisms13020251

**Published:** 2025-01-24

**Authors:** Marcela Ewert, Brook L. Nunn, Erin Firth, Karen Junge

**Affiliations:** 1Polar Science Center, Applied Physics Laboratory, University of Washington, 1013 NE 40th Street, Box 355640, Seattle, WA 98105-6698, USA; marcela.ewert@edmonds.edu (M.E.); pontus@uw.edu (E.F.); 2Department of Genome Sciences, University of Washington, Foege Building S-250, Box 355065, 3720 15th Ave NE, Seattle, WA 98195-5065, USA; brookh@uw.edu

**Keywords:** halophiles, psychrophiles, *Colwellia*, *Halomonas*, *Psychrobacter*, proteomics

## Abstract

Few halophilic strains have been examined in detail for their culturability and metabolic activity at subzero temperatures, within the ice matrix, over the longer term. Here, we examine three Arctic strains with varied salinity tolerances: *Colwellia psychrerythraea* str. 34H (Cp34H), *Psychrobacter* sp. str. 7E (P7E), and *Halomonas* sp. str. 3E (H3E). As a proxy for biosignatures, we examine observable cells, metabolic activity, and recoverability on 12-month incubations at −5, −10 and −36 °C. To further develop life-detection strategies, we also study the short-term tracking of new protein synthesis on Cp34H at −5 °C for the first time, using isotopically labeled ^13^C_6_-leucine and mass spectrometry-based proteomics. All three bacterial species remained metabolically active after 12 months at −5 °C, while recoverability varied greatly among strains. At −10 and −36 °C, metabolic activity was drastically reduced and recoverability patterns were strain-specific. Cells were observable at high numbers in all treatments, validating their potential as biosignatures. Newly synthesized proteins were detectable and identifiable after one hour of incubation. Proteins prioritized for synthesis with the provided substrate are involved in motility, protein synthesis, and in nitrogen and carbohydrate metabolism, with an emphasis on structural proteins, enzymatic activities in central metabolic pathways, and regulatory functions.

## 1. Introduction

Important targets in the search for life in the solar system are expected to be cold and saline, with accessible biosignatures most likely entrapped in ice. For instance, large and long-lived saline oceans underlay the kilometers-thick ice covers of Enceladus [1,2], Europa [3,4], and possibly even Pluto [5]; likewise, the polar ice caps in Mars may harbor a layer of perchlorate brine at its base [6]. These icy worlds are a current priority of space exploration. The Europa Clipper, a mission to further explore Europa’s habitability, has already launched, with a projected arrival date of 2030 [7]. Concepts for a Europa lander mission, which would target the icy surface, already exist [8]. The Mars 2020 mission, with its accompanying Perseverance rover, is already carrying out an in situ investigation of ancient-life biosignatures on the surface of Mars, with the prospect of collecting samples to return to Earth [9]. Understanding the habitability of cold and saline environments in the solar system, including the challenges associated with retrieving and processing samples, can be informed by Earth’s own cold and saline environments and the microorganisms that inhabit them.

Earth’s polar environments have been extensively studied for their cold-active and phylogenetically diverse extremophiles, which thrive in brine-rich sea-ice matrices, as well as glacial veins and inclusions [10,11]. The last two decades of research have greatly increased our understanding of low-temperature biology, suggesting the possibility of microbial life on icy worlds [10]. Methods include the incorporation of C14- or 3H-labeled radioisotopes [12,13,14,15,16,17], occasionally ATP pools [18] and, more recently, stable isotopes [19,20]. The lower temperature limits for life in icy environments currently stand at −15 °C for bacterial growth [16], −20 °C for protein and DNA synthesis [21], −33 °C for respiration in heterotrophic organisms [14] and −35 °C for photosynthesis in Antarctic lichens [22]. These temperatures fall within the expected range at the bottom of Mars’ southern ice cap, assumed to vary between −103 °C and −3 °C [6]. Temperature limits for microbial activity have been studied with isolates obtained from a variety of Earth’s polar habitats, including *Paenisporosarcina* sp. B5 and *Chryseobacterium* sp. V3519-10 from Antarctic glacial basal ice [14]; *Colwellia psychrerythraea* str. 34H from Greenland sediments [23]; *Planococcus halocryophilus* str. Or1 from Arctic permafrost [16]; *Psychrobacter* sp. str. Trans1, *Arthrobacter* sp. str. G200-C1 from glacial ice [15]; and the lichen *Umbilicaria decussata* from ice-free areas in maritime Antarctica [22]. Temperature, though, is not the only factor relevant to consider potential microbial life in icy worlds. Extraterrestrial environments may have dominant salt profiles that differ from those on Earth, such as the perchlorate-rich brines on Mars [24]. Organisms known to tolerate simulated Martian conditions, such as the cold-adapted ecological generalist *Serratia liquefaciens* or *Colwellia psychrerythraea* st. 34H, have also been studied [20,25].

Current and planned missions to icy worlds only have the ability to observe biosignatures at the surface of icy moons and planets. Microorganisms (or their fragments) existing at deeper, more hospitable regions, though, could be entrapped in ice and then be transported to the surface by geological processes [26]. As such, from an astrobiological perspective, it is of interest to investigate the survival and metabolic activity of microorganisms during long-term entrapment in ice.

Psychrophilic and halophilic isolates are rarely studied within the icy matrix itself, and most of the existing subzero microbial activity studies have been carried out over the course of a few months at most (for exceptions, see [23,27]). Yet, life detection on icy planetary bodies requires us to recognize and interpret the morphological and metabolic features left by microbes entrapped in ice for extended periods of time. Growth and reproduction is considered a key feature diagnostic of life [28]; it may involve the observation of preserved cell-like structures or even the recovery of cells after long-term entrapment in the ice—as has been proved feasible in terrestrial glacial ice [29]. Recoverability experiments can also be important in the context of sample–return missions, as they could inform on the best practices to maintain samples containing potential life forms until their return to Earth.

Aside from methods requiring microscopic observation or cultivation, the existence of ongoing metabolism could also be indicative of life [28]. Earth’s life shares a common carbon-based chemistry involving the use of amino acids as building blocks [30]. While it is unknown if extraterrestrial life would share the same chemistry as Earth’s life, it is expected that amino acids and amino-acid precursors were present throughout the solar system in its early history [30], and as such, could be part of life elsewhere. The incorporation of a radioactively labeled amino acid could be thus used as a biosignature, assuming live microorganisms can still show activity after being subjected to low temperature and high salinity for long periods of time.

Our research aims to understand how long-term entrapment in different subzero environments can impact microbial survival, their longevity, and the retrieval of key biosignatures such as metabolic activity, recoverability, and cell abundance. Data on the activity and recoverability of microorganisms after long-term entrapment in ice is sparse for halophilic organisms. However, we expect that species differing in their adaptation to salinity and temperature will react in unique ways. Our experiments examine 12-month subzero incubations in the seawater of three marine strains with different tolerances for temperature and salinity: *Colwellia psychrerythraea* str. 34H (Cp34H), *Psychrobacter* sp. str. 7E (P7E), and *Halomonas* sp. str. 3E (H3E). These isolates were either directly obtained from sea-ice brines (P7E and H3E [31,32]) or found to be present therein (Cp34H [33]). Cp34H is an obligately psychrophilic strain with a temperature growth range of −12 to 19 °C. Cp34H is the most susceptible to high salinities, with a growth range of 15–60 ppt (50% to 200% of sea salt concentration), but still shows signs of metabolic activity up to 140 ppt. On the contrary, P7E and H3E are psychrotolerant and can grow at temperatures between −8 to 25 °C and −1 to 25 °C, respectively. These two strains also have wider tolerances for salinity: P7E can grow in salinities between 17 and 125 ppt, and H3E, the most salinity-tolerant of the three strains, has growth confirmed for salinities between 35 and 220 ppt (lower salinities were not tested). Salinity and temperature tolerances for Cp34H were previously published in the literature [34,35,36,37], while those for P7E and H3E were determined in previous experiments (Appendix A).

Here, we report the results of incubations at three temperatures associated with distinct frozen environments. −5 °C is a common temperature encountered in sea ice, characterized by the low salinity and high connectivity of brine channels [38]. At −10 °C, brine salinity is higher and the connectivity of sea ice is much lower, limiting the exchange of nutrients and metabolic waste. Finally, the incubations at −36 °C represent the lowest temperature on Earth where liquid water would be expected, based on the equilibrium freezing pathway of seawater [39,40]. In addition to cell numbers, recoverability and activity, we report on the feasibility of using isotopically labeled ^13^C-leucine and mass spectrometry-based proteomics as a biosignature-detection method. Stable isotope probing (SIP) is a non-radioactive technique that can demonstrate metabolic and molecular level dynamics (e.g., [41,42]). SIP is commonly used at temperatures above 0 °C, although there are a few low temperature examples [19,21]. When used in conjunction with proteomics (protein-SIP), this method can retrieve detailed information about specific metabolic pathways and track short-term-specific new proteins as they are synthesized through time [43]. Protein-SIP employs heavy isotopes such as ^15^N, ^18^O and ^13^C [44]. To the best of our knowledge, to date, this technique has not been used to study subzero microbial activity and associated molecular signatures. Here, we demonstrate that using ^13^C-leucine can track the synthesis of new proteins at subzero temperatures (−5 °C) for our psychrophilic strain (Cp34H). Proteins synthesized under long-term subzero incubations could provide key targets for future in situ environmental or off-planet experiments on icy worlds.

## 2. Materials and Methods

### 2.1. Cell Cultures and Incubations

#### 2.1.1. Cell Cultures

Three Arctic strains, *Colwellia psychrerythraea* str. 34H (Cp34H) [ATCC:27364; NCBI:txid28229], *Psychrobacter* sp. str. P7E (P7E) [NCBI:txid2058322], and *Halomonas* sp. str. H3E (H3E) [NCBI:txid2058321], were grown from frozen glycerol stocks provided by the Deming laboratory (University of Washington, Seattle, WA, USA). All strains were grown at −1 °C in half-strength Marine Broth 2216 (Difco laboratories, Detroit, MI, USA) with periodic mixing (P7E, H3E) until late-log (Cp34H) or early-stationary phase (P7E, H3E). Growth was assessed by optical density (OD600) measurements and/or cell counts (as described in Section 2.2). Cell densities at the time of harvest were 1.01 × 10^9^ cells/mL for Cp34H; 3.6 × 10^8^ cells/mL for P7E, and 8.29 × 10^7^ cells mL for H3E.

#### 2.1.2. Harvest

Supernatant-free cells were harvested by a process of serial centrifugation [37]. Briefly, cultures were centrifuged at 2800× *g* for 20 min at 4 °C to pellet cells and discard the supernatant. Pelleted cells were resuspended in pre-chilled (4 °C) autoclave-sterilized artificial seawater (ASW) [411 mM NaCl, 9.4 mM KCl, 26.3 mM MgCl_2_, 28.4 mM MgSO_4_*7H_2_O, 5 µM TAPSO (3-[N-tris(hydroxymethyl) methylamino]-2-hydroxypro-pane-sulfonic acid); pH 7.6] and centrifuged twice again (2800× *g*, 10 min, 4 °C). Between each centrifugation round, the supernatant was discarded and the cells were resuspended in pre-chilled ASW. For H3E, the initial centrifugation step had to be repeated twice and at a higher force (3026× *g* for 20 min, 4 °C), due to the high viscosity of the supernatant.

#### 2.1.3. Resuspensions

To ensure similar cell densities for all treatments, resuspensions were sub-sampled to perform cell counts (see Section 2.2) and then diluted with sterile, pre-chilled ASW to a target concentration of 10^7^ to 10^8^ cells/mL. Diluted resuspensions were then amended with nutrients and vitamins to a final concentration of glucose (1 g/L), leucine (4 µg/L), thymidine (6 µg/L), and vitamins (5 mL/L 100 × RPMI 1640 Vitamin Mix [Sigma-Aldrich, St. Louis, MO, USA]). Nutrients were added to prevent cell starvation and were chosen based on growth studies for Cp34H [45,46]. After nutrient additions, cells were allowed to acclimate to the new media overnight at −1 °C. Cell counts performed after acclimation resulted in Cp34H: 2.52 × 10^8^ ± 6.03 × 10^7^ cells/mL, P7E: 7.48 × 10^7^ ± 2.01 × 10^7^ cells/mL, and H3E: 2.06 × 10^7^ ± 8.85 × 10^6^ cells/mL (Mean ± SD; *n* = 12). For samples measuring radioisotope and stable isotope incorporation, 4 µg/L [^3^H]-leucine and 1.4 mg/L [^13^C_6_]-leucine (Cambridge Isotope Labs, Cambridge, MA, USA), respectively, were used and added at experimental time point T_0_ (see Section 2.3 for more details).

#### 2.1.4. Incubations

Cell resuspensions from each of the three strains were incubated at one control temperature and three subzero temperatures. The control temperature was the optimal growth temperature for Cp34H (8 °C [35]) and close to the fastest growth temperature known for P7E and H3E (22 °C, see Appendix A). Incubations at the control temperature were kept only for one month, given the rapid growth of the organisms. Subzero incubations were performed at −5 °C, −10 °C, and −36 °C, with samples collected throughout the 12-month incubation. For cell abundance and recoverability, samples were collected at the following time points: 0 h, 1, 7, 14, and 28 days, and 2, 6, and 12 months. For [^3^H]-leucine incorporation, samples were collected more frequently during the first day (0, 1, 2, 4, 8, 12, 24 h) and subsequently in the same pattern as that of cell parameters (7, 14, 28 days, and 2, 6, and 12 months). For protein-SIP, samples were collected during the first week only, in the same pattern as that of [^3^H]-leucine incorporation (0, 1, 2, 4, 8, 12, 24 h, and 7 days). Given the amount of samples, incubations for each strain were setup at separate times, two months apart from each other. A detailed timeline is available in Appendix A.

### 2.2. Cell Abundance and Recoverability

Cell abundance and recoverability (defined here as the ability to culture bacterial cells from a sample) were evaluated for each sample following methods described before [37]. To measure recoverability, 20 µL triplicates of each sample were serially diluted (1:10 dilution) ten times into half-strength Marine Broth 2216 in 96-well plates, which were then incubated at control temperature for one week (8 °C for Cp34H, 22 °C for P7E and H3E). Recoverability was calculated from the number of wells showing growth according to standard tables from the FDA Bacteriological Analytical Manual [47]. To measure cell abundance, the remainder of sample volume was fixed in 2% formaldehyde, stained with the DNA-fluorescent stain DAPI (4′,6′-diamidino-2-phenylindole dihydrochloride) and enumerated via epifluorescent microscopy (as in [48]).

### 2.3. Metabolic Activity

#### 2.3.1. [^3^H]-Leucine Incorporation

The incorporation of [^3^H]-leucine indicates protein synthesis, a measurement of metabolic activity. Radioisotope solutions were created by diluting the radioisotope stock 1:100 in sterile ASW. Two different stocks were used in these experiments. For Cp34H and P7E, we used [^3^H]-leucine: 54.1 Ci mmol^−1^ (MP Biomedicals, Santa Ana, CA, USA, Catalogue No. 0120036E01). For H3E we used leucine, L-[4,5-^3^H]: 149 Ci mmol^−1^ (Perkin Elmer, Springfield, IL, USA, Catalogue No. NET1166001MC, in a sterile 2:98 ethanol–water mixture). To account for the higher radioactivity of the 149 Ci mmol^−1^ radioisotope, the leucine incorporation readings (DPM) for H3E were adjusted by dividing them by 2.754. Cell suspensions amended with the radioisotope solution were sampled for [^3^H]-leucine incorporation following methods developed for Cp34H in saline ice [23,37,49].

To amend samples, we added 100 µL of the diluted [^3^H]-leucine radioisotope solution to triplicate 500 µL aliquots of the cell resuspensions (“live” samples). Triplicate killed controls were created by adding 100 µL 50% chilled trichloroacetic acid (TCA) solution prior to radioisotope tracer addition. For all samples, two ethanol washes were performed during the processing stage [37]. Samples were assayed for radioactivity (dpm) using a Tri-Carb 2800TR liquid scintillation analyzer (Packard Bioscience Company, Meriden, CT, USA) as in [49]. The final values for [^3^H]-leucine incorporation correspond to the live measurements minus the average of the three killed controls.

#### 2.3.2. Growth Rates

Protein production rates were calculated from linear time portions of [3H]-leucine incorporation values and scaled to bacterial carbon mass according to established methods [50]. Common conversion factors and values were used to enable a cross-comparison of literature values (131.2 g mol^−1^ = molecular weight of leucine; 7.3 mol% = fraction of leucine in protein; 0.86 = ratio of cellular carbon to protein [51]; 65 fg C bacterium^−1^ [49,52]). Growth rates, calculated from rates of leucine incorporation, are presented in comparison with those reported or derived from the literature for other organisms across a range of temperatures (see Appendix B and Appendix A).

### 2.4. Stable Isotope Probing (SIP) and Proteomics

To determine newly synthesized proteins over time, we focused on Cp34H, the only one from our strains with an available, high-quality genome sequence. Five replicates of Cp34H cell suspensions, per time point, were incubated with 1.4 ppm of ^13^C_6_-leucine at −5 °C. Cells were harvested throughout the first week as follows. At each time point, incubations were stopped by adding equal volumes of 50% TCA, for a final concentration of 25% TCA. After TCA addition, cells were pelleted by centrifugation (18,000× *g* for 15 min, 4 °C) and the supernatant was removed. Samples were stored at −80 °C until further processing. Two batches of cells were harvested at T = 0 h; in one case, TCA was added after the addition of ^13^C_6_-leucine (representing T = 0 live), and in the other case, TCA was added before the addition of ^13^C_6_-leucine, representing the killed control for the experiment.

#### 2.4.1. Sample Preparation

Harvested cell pellets were gently rinsed with 50 µL 1× phosphate-buffer solution and then lysed in 5% SDS, 50 mM triethylammonium bicarbonate (TEAB), 2 mM MgCl_2_, 1× HALT protease, and phosphatase inhibitors, using a titanium micro-probe sonicator (Branson 250 Sonifier, Sonitek, Milford, CT, USA; 20 kHz, 10 times for 10 s on ice). Protein concentrations were measured in triplicate using the bicinchoninic acid micro-assay (BCA, ThermoScientific, Norristown, PA, USA) with bovine serum albumin (BSA) standard. A 20 μg aliquot was collected from each sample, to which 320 ng of enolase was added as an internal standard and benzonase (62 units) was added to degrade DNA. Protein disulfide bonds were reduced by dithiolthreitol (20 mM, 10 min, 95 °C) and alkylation was completed using iodoacetamide (40 mM, 30 min, in dark, 20 °C). Following alkylation, the sample was acidified with 12% phosphoric acid (1:10 ratio with sample volume). To each protein lysate, 350 μL S-Trap binding buffer (100 mM TEAB in 90% methanol) was added. ProtiFi S-Trap spin columns [53] were used for on-column digestions and protein sample clean-up. Protein digestion was completed using Promega Trypsin, Promega Corporation, Madison, WI, USA (2 μg in 50 mM TEAB). Digested proteins (peptides) were eluted with 50% of acetonitrile containing 0.2% of formic acid. Peptide samples were then dried down, resuspended in 2% acetonitrile with 0.1% formic acid, and frozen at −80 °C until liquid chromatography–tandem mass spectrometry (LC-MS/MS) analyses were performed.

#### 2.4.2. Mass Spectrometry

For each LC-MS/MS analysis, 1 μg of total protein was sampled, injected, and analyzed using a Thermo Fisher (San Jose, CA, USA) QExactive (QE). Samples were separated and introduced into the mass spectrometer (MS) by reverse-phase chromatography using a 40 cm long, 75 μm i.d., fused silica capillary column packed with C18 particles (Magic C18AQ, 100 Å, 5; Michrom, Bioresources, Auburn, CA, USA) and fitted with a 3 cm long, 100-μm i.d. precolumn (Magic C18AQ, 200 Å, 5; Michrom). Peptides were eluted using an acidified (formic acid, 0.1% *v*/*v*) water–acetonitrile gradient (2–35% acetonitrile in 90 min). The mass spectrometer was operated in data-dependent acquisition (DDA) mode, where the top 20 most intense ions were selected for MS2 acquisition from MS1 scans of 400–1600 *m*/*z*. MS1 resolution was set to 70,000 with an AGC target of 1E6 and a maximum IT fill time of 100 ms. MS2 mode was operated with a resolution of 35,000, AGC target of 5e4, maximum IT fill time of 50 ms, and isolation window of 1.2 *m*/*z*. DDA settings included the charge exclusion of unassigned, +1, and ≥ +6 ions, with a 10 s dynamic exclusion. Quality control (QC) peptide mixtures were analyzed every sixth injection to monitor chromatography and MS sensitivity and all standard peptides were required to maintain a CV < 20%.

#### 2.4.3. MS Data Analysis

MS data analysis was carried out as described before [37]. In brief, all the MS/MS generated spectra were searched and interpreted using COMET [54,55]. The protein database used for correlating spectra with protein identifications was generated by combining the Uniprot *Colwellia psychrerythraea* protein database consisting of 4910 proteins (downloaded 4 February 2019) concatenated with 50 common contaminants and the quality control peptides. COMET parameters included reverse concatenated sequence database search, trypsin enzyme specificity, 10 ppm MS1 peptide mass tolerance, fixed cysteine modification of 57 Da (resulting from the iodoacetamide), and the variable modifications on methionine of 15.999 Da (oxidation) and leucine of 6.02 Da (^13^C_6_-leucine incorporation). Concatenated target–decoy databases searches were completed. To confidently identify stable isotope labeled peptides observed at each time point, peptide spectra were filtered at probability 0.99 using the trans-proteomic pipeline (TPP; *.pep.xml; [56]). All data were uploaded to the Limelight Server [57], available via ProteomeXchange with the identifier PXD059470, as described in the Data Availability Statement. The final list of ^13^C_6_-leucine proteins only includes those identified with ≥3 unique peptide spectra from all of the replicates.

## 3. Results

### 3.1. Temperature-Dependent Response in Cell Abundance, Recoverability and Activity

#### 3.1.1. Control Temperature

All three strains maintained cell numbers similar to those initially inoculated (Figure 1a). Contrary to cell abundance, cell recoverability showed marked differences among strains, with P7E being the most affected (Figure 1b). At the start of the experiment, the recoverability of P7E began at 0.5–1.2 × 10^7^ recoverable cells/mL, similar to that of the other strains. By the end of the 1-month observation period, though, there was only one P7E replicate with recoverable cells, and its recoverability was 7 orders of magnitude lower than the initial. Cp34H lost 1 to 2 orders of magnitude in recoverability, from 1.2–2.5 × 10^8^ cells/mL at the start point to 0.2–7.5 × 10^6^ cells/mL after 1 month. The only strain able to preserve its initial recoverability was H3E, with losses of less than an order of magnitude of recoverable cells throughout the 1-month period.

The incorporation of [^3^H]-leucine indicated that all strains exhibited metabolic activity, reaching maximum [^3^H]-leucine incorporation within the first 24 h of incubation (Figure 1c). This maximum incorporation was of similar magnitude for all strains: Cp34H (1.8–2.3 × 10^5^ DPM at 4 h), P7E (2.2–2.4 × 10^5^ DPM at 8 h) and H3E (1.6–1.7 × 10^5^ at 1 d). After the first day, leucine incorporation for Cp34H and H3E decreased, reaching the lowest values after 1 month (3.0–5.6 × 10^4^ DPM for Cp34H and 0.91 –1.1 × 10^5^ DPM for H3E). On the contrary, P7E showed a secondary increase in leucine incorporation after the first week, resulting in a second peak of leucine incorporation after a month (2.1–2.2 × 10^5^ DPM). Overall, the lowest [^3^H]-leucine uptake occurred in CP34H and the highest in P7E.

#### 3.1.2. Incubations at −5 °C

After a year of incubation at −5 °C, all strains maintained cell numbers close to those initially inoculated (gray circles in Figure 2). The only noticeable change was observed in Cp34H, which had a decrease of 1 order of magnitude in cell abundance, from 2.3–2.5 × 10^8^ cells/mL at the start of the incubation to 2.0–4.2 × 10^7^ cells/mL at the 1-year end point. Recoverability, though, varied much more with time, with each strain displaying unique behavior (colored circles in Figure 2). At −5 °C, only H3E maintained recoverability throughout the 1-year incubation. Cp34H recoverability dropped rapidly, losing 7 orders of magnitude in 6 months, from 2.1–7.5 × 10^8^ cells/mL at the time of incubation to 18–46 cells/mL after 6 months, to having no recoverable cells observed in most replicas after 1 year of incubation. P7E was the most susceptible organism, not having any recoverable cells after 6 months of incubation at −5 °C. Appendix A shows a more detailed view of the loss of recoverability for P7E in the first two months.

All strains were metabolically active at −5 °C, with [^3^H]-leucine uptake being lowest in Cp34H and highest in P7E (Figure 3). The maximum levels of leucine incorporation for all three species were on the same order of magnitude than those at control temperature. All three species had a peak of leucine incorporation during the first day, with Cp34H and P7E having a secondary increase later in the incubation. Cp34H and P7E both had a gradual increase in activity in the first 6 months, plateauing between 6 months and 1 year. On the contrary, after the initial peak, leucine incorporation for H3E remained at a similar level throughout the experiment. The highest leucine incorporation occurred at 1 year for Cp34H (1.2–1.7 × 10^5^ DPM), at 6 months for P7E (2.0–3.0 × 10^5^ DPM), and at day 1 for H3E (1.6–1.7 × 10^5^ DPM). For a detailed view of leucine incorporation during the first 24 h, see Appendix A.

#### 3.1.3. Incubations at −10 °C

Following a similar trend to that observed at −5 °C, cell numbers after a year of incubation were close to those initially inoculated for all strains (gray circles in Figure 2). The cell abundance of Cp34H decreased one order of magnitude, from 2.6–3.2 × 10^8^ cells/mL at the start of the incubation to 3.6–4.0 × 10^7^ cells/mL at the 1-year end point. P7E increased from 4.7–7.6 × 10^7^ cells/mL at the start of the incubation to 1.1–1.5 × 10^8^ cells/mL at the 1-year end point. Likewise, H3E doubled from 2.1–3.8 × 10^7^ cells/mL to 4.2–4.8 × 10^7^ cells/mL at the 1-year end point. The ability to maintain cell recoverability was strain-specific (the colored circles in Figure 2). H3E was the only strain that maintained cell recoverability after a year of incubation. For Cp34H and P7E, recoverability was quickly lost after a month of incubation, showing a faster decline than what was observed at −5 °C (Figure 2).

Leucine incorporation in all three strains was strongly reduced at −10 °C compared with −5 °C (Figure 3; note the difference in axes). Notably, for H3E, no significant metabolic activity was detectable in most replicas. The majority of the samples had almost no incorporation above that of the killed controls, and the activity was below or equal to that of the t = 0 (see Appendix A for a detailed view of leucine incubation during the first day). The maximum rates of leucine incorporation for Cp34H and P7E were between one and two orders of magnitude lower than those observed at −5 °C, but still showed a bimodal distribution, with a peak in the first 24 h and a slow increase later in the incubation. For Cp34H, the maximum rate occurred at 4 h (5.8–7.3 × 10^3^ DPM), while for P7E it occurred after a year (1.2–2.2 × 10^3^ DPM—note that P7E had one replica with a high peak of 3.0 × 10^3^ DPM at 12 h, but there was wide variation among replicas at the same time point).

#### 3.1.4. Incubations at −36 °C

Each strain had a different pattern of cell abundance at the lowest incubation temperature (Figure 2). For Cp34H, cell abundance remained constant for the first six months, then declined slightly in a less pronounced manner than it did at −5 °C or −10 °C (from 2.3–3.5 × 10^8^ cells/mL at the starting point to 1.4–2.3 × 10^8^ cells/mL at the one-year mark). Cell abundances for P7E behaved similarly than at other subzero temperatures, with only slight changes throughout the experiment. H3E showed a pattern of increasing cell numbers, from 0.8–1.7 × 10^7^ cells/mL at the starting point to 3.7–6.2 × 10^7^ cells/mL after a year. The recoverability of all three strains was markedly different at −36 °C compared to −10 °C (Figure 2). For Cp34H and P7E, this lower temperature had a protective effect. Contrary to what was observed at −10 °C, incubations at −36 °C resulted in the presence of recoverable cells for Cp34H (20 to 115 recoverable cells/mL) and notably for P7E (2.1–7.5 × 10^5^ recoverable cells/mL) at the 1-year end point. For H3E, this was the only temperature for which recoverability decreased over time, from 0.2–1.2 × 10^8^ recoverable cells/mL at the starting point to 0.2–2.2 × 10^5^ recoverable cells/mL after a year. Overall, Cp34H appeared the most susceptible to a decrease in recoverability at this temperature.

Leucine incorporation at −36 °C was comparable to that observed at −10 °C for all three strains, both in magnitude and trends (Figure 3). H3E continued showing no significant metabolic activity, with very few samples detecting leucine incorporation above that of killed controls. Cp34H and P7E presented a bimodal distribution, with a peak during the first 24 h and a secondary peak later in the incubation. For Cp34H, the maximum rate occurred at 1 year (3.5–6.1 × 10^3^ DPM), while for P7E it occurred in the first hour (1.0–1.3 × 10^3^ DPM). It is notable that, for Cp34H, the rate of leucine incorporation at 1 year was the highest value recorded at any point among the three bacterial strains at either −10 °C or −36 °C.

### 3.2. Stable Isotope ^13^C_6_-Leucine Incorporation at −5 °C in Cp34H

The incorporation of ^13^C-labeled leucine into newly synthesized proteins was observed in all live Cp34H samples at −5 °C. The spectrophotometric BCA assay showed an overall increase in protein concentration in the samples over time (Figure 4a), consistent with an increase in the number of labeled peptides/proteins (Figure 4b). Significantly labeled proteins (those with at least three peptide spectrum matches (PSMs) containing the ^13^C_6_-leucine label) increased in both abundance and diversity over time (Figure 4b). Flagellin was the first significantly labeled protein, detected at 1 h of incubation when it reached three PSMs with a 0.99 probability. Its abundance continued to rise throughout the experiment, peaking at the 7-day endpoint. This is clearly shown in the sharp increase in PSM counts for flagellin compared to other proteins. Proteins involved in fatty acid degradation and leucine metabolism became significantly labeled from the 2 h time point, with their labeling continuing through to the 7-day mark (168 h). These proteins showed a steady increase in PSM counts over time, indicating ongoing synthesis. In addition to these pathways, proteins related to central metabolism, nitrogen metabolism, fatty acid metabolism, transportation, and stress response became significantly labeled at varying time points. Although their PSM counts were generally lower than flagellin, they still demonstrated a gradual increase, particularly between the 12 h and 7-day timepoints, marking a delayed but significant incorporation of ^13^C_6_-leucine.

The identification of significant labeling in more diverse proteins over time suggests a broadening cellular response to experimental conditions. Further specifics on significantly labeled proteins at each time point are available in Appendix A.

## 4. Discussion

### 4.1. Strain-Specific Biosignatures

The exploration of remote Earth’s subzero environments, such as the polar regions, happens far less frequently compared to their temperate counterparts. Understanding ecosystem processes in these environments can be informed by the behavior of observable, recoverable, and metabolically active cells exposed to subzero temperatures for long periods of time. Since cellular structures, growth, and metabolic activity are considered biosignatures, these experiments can also inform expectations for the search for life in extraterrestrial environments dominated by ice, such as Jupiter’s moon Europa.

Icy moons such as Europa could have habitats in the ice shell, the water column, or the benthos (see Table 2 in [26]). The thickness of the ice shell, paired with limitations in current technologies, imply that biosignatures from these environments could only be observable after being transported to the surface of the ice. As such, biosignature detection on icy worlds assumes that microorganisms or biological molecules from habitats below the surface can be entrapped in ice and then transported to the surface by geological activity or impacts [26]. Some of the geological processes transporting biosignatures, such as cryovolcanic activity, could involve rapid freezing. Once transported to the surface of the ice shell, biosignatures would be exposed to very low temperatures for extended periods of time.

Darwinian evolution, one of the main properties of life [28], makes it reasonable to expect that if life exists in extraterrestrial settings, more than one species of organism would be present. In Earth’s own cryosphere, a diverse set of microbial groups survive and remain active at subzero temperatures due to their varied adaptations to the low temperatures and high solute concentrations associated with the freezing process [10,58,59]. As such, to better understand biosignatures produced by complex microbial communities, it is necessary to consider that species-specific responses to long-term entrapment in ice may result in different observable signs of life. In this work, we evaluated three genera with varied tolerances to low temperature and high salinity, with each strain exhibiting a unique response to the incubation conditions.

Our long-term incubations in minimal media demonstrated that cells stand out as a reliable biosignature in frozen environments, especially at the lowest temperatures. All three strains had microscopically observable cells after a year of low-temperature incubations, with end-points for cell numbers similar to those initially inoculated. Cell numbers were mostly maintained for the two strains originally isolated from icy environments (P7E and H3E). Cp34H was the only strain to experience measurable cell loss, in particular at −5 °C and −10 °C, where end-point cell numbers were one order of magnitude lower than those initially inoculated. However, even for Cp34H, a large number of observable cells remained.

Some of the cells remaining after the 1-year incubation were recoverable and viable. H3E was the most resilient strain from those evaluated, maintaining a high number of culturable cells at all tested temperatures after 12 months of incubation. The presence of recoverable cells after long-term freezing is consistent with results from the environmental sampling of glaciers [29]; based on these results, the potential for finding recoverable organisms after long-term freezing should not be discarded. H3E has previously not been studied, and we suggest that an in-depth exploration of its life processes at lower temperatures is a promising avenue for research. The *Halomonas* genus is widely known for its presence in a variety of saline environments [60,61], including hypersaline subzero environments in the Canadian Arctic [62], where it was reported to grow at −5 °C.

Long-term freezing at moderately low subzero temperatures, though, strongly affected the recoverability of cells for the other two strains. For instance, Cp34H and P7E maintained a very low number of recoverable cells after a year of incubation at −5 °C and −10 °C. In some cases, no culturable cells were identified. Recoverability improved for both Cp34H and P7E at −36 °C, indicating that the lowest temperature tested here had a protective effect. Temperature-driven differences in recoverability are possibly driven by the process of freezing and will be discussed in the next section. Different culturing methods may also be necessary, as some organisms might require a recovery period under low-nutrient conditions while they repair cellular damage and detoxify from metabolic byproducts before they can reproduce in high-nutrient media [29].

The variations in recoverability and activity of each strain highlight the need to test different parameters when searching for life; focusing on only one parameter may miss microorganisms with dissimilar responses. For instance, while H3E maintained a high number of recoverable cells at −10 °C, significant long-term metabolic activity was not observable. P7E on the other hand, did not have recoverable cells at −10 °C with the culturing methods used, but was found to be metabolically active. Cells more metabolically active during freezing (such as P7E and Cp34H) may have more metabolic byproducts and thus, need more time or more specific low-nutrient media to detoxify before recovery. Studying the relationship between metabolic activity and recovery after long-term entrapment in ice is an intriguing avenue of research; it could improve methods of biosignature search by indicating whether recovery efforts would benefit from reduced nutrient media or longer periods of incubation to facilitate detoxification and repair. Strains such as P7E, which maintained its metabolic activity even at −36 °C for the 2- and 6-month time points, underscore it as a potential candidate to further elucidate strategies for microbial metabolic activity and recovery under extreme subzero conditions [59].

### 4.2. The Frozen Environment

Once freezing happens, organisms are confined to increasingly smaller liquid pockets in the ice and experience increased osmotic stress [10]. In the laboratory setting, however, freezing does not always occur at the expected freezing point. While our experimental setup showed consistent freezing in treatments at −10 °C and −36 °C, samples incubated at −5 °C remained liquid (supercooled). Equations for the freezing of seawater [63] indicate that our −10 °C and −36 °C treatments exposed microbes to salinities of 143 and 289, respectively, while the −5 °C treatment exposed microorganisms only to the stress of low temperature without the added stress of high salinity.

In our experimental setup, freezing did not greatly affect the long-term permanence of cells observed microscopically by DAPI stain. All strains maintained similar cell numbers throughout the experiment when comparing the −5 °C (supercooled) and −10 °C (frozen) treatments. This is an encouraging result, highlighting the possibility of identifying cell structures as biosignatures on frozen environments.

As for recoverability of the observable cells, the freezing process had a large effect on recoverability for Cp34H only. When kept at −5 °C, 34H was able to maintain high levels of recoverability during the first two months. In contrast, recoverability dropped rapidly at −10 °C, with almost no culturable organisms present after 2 months of incubation. The −10 °C incubation temperature is within the growth range for Cp34H (down to −12 °C [36]), but the expected brine salinity at this temperature greatly exceeds Cp34H growth range (no growth above 60 ppt [35]). We hypothesize that the high salinity of the −10 °C brines could be one factor responsible for the limited recoverability of this strain. On the contrary, the other two strains, known for their ability to grow at higher salinities, maintained similar patterns of recoverability at −5 and −10 °C, indicating that the increased salinity does not pose additional stress on the cells.

The most salient difference between the −5 and −10 °C treatments was a drastic decrease in activity for all three strains. Two possible factors could drive this change in metabolic activity. On one hand, the combination of high salinity and low temperature could stress the cells to the point of triggering a reduction in activity, so that only the most basic maintenance functions occur. On the other hand, our samples could have undergone vitrification, a process by which liquids solidify without crystallization, directly transforming from a liquid state to a glass, ice-free state. Vitrification has been found to be favorable to the preservation of viability, while at the same time associated with a reduction in metabolism [64]. While vitrification is usually associated with the lowest temperatures, it can occur at temperatures as high as −10 °C [65].

Salinity alone does not explain the reduction in activity. While increased brine salinity at −10 °C is above the growth range for 34H and P7E, the most extreme decrease in activity was seen in H3E, a species much more tolerant of high salinity. H3E showed no significant metabolic activity, with almost no leucine incorporation detected at −10 °C or above that killed controls or t = 0 measurements. Since the brine salinity at −10 °C is within the growth range for H3E, vitrification could be playing a role. In a vitrified state, the high internal viscosity of the cell slows down the diffusion of oxygen and consequently, the metabolic activity, to the point of ceasing cellular metabolism [65]. A vitrified state would be consistent with the high recoverability of cells experienced by H3E and the increased survival of recoverable cells at −36 °C for Cp34H and P7E.

### 4.3. Metabolic Activity and Generation Times

In this study, we used leucine incorporation to predict if organisms are carrying out maintenance functions or growth [52]. Our results at −5 °C indicate maintenance metabolism for all three strains, with maximum rates of protein synthesis ranging from 1.0–9.2 × 10^−4^ gC gC^−1^ h^−1^ over the course of the experiment. The transition from −5 °C to −10 °C has a large impact on microbial activity for all strains, as discussed above, likely a result of freezing. This indicates a large reduction in protein synthesis and a cessation in growth metabolism. Maximum rates of protein synthesis at −10 °C were 2.0–9.1 × 10^−6^ gC gC^−1^ h^−1^ for Cp34H and P7E, also within ranges associated with maintenance metabolism. Almost no protein synthesis, though, was detected for H3E. Comparisons of protein synthesis involving leucine markers, though, need to consider that aliphatic amino acids (such as leucine) can hinder the flexibility of proteins at low temperatures, and as such, are under-represented in cold-adapted proteins [66].

Generation times were obtained from values of metabolic activity and compared with published values for a variety of organisms, growth conditions, and temperatures (Figure 5). Our organisms exhibited a similar pattern to those published in the literature. As expected, generation time peaked around the optimal temperature. As temperature decreased to −5 °C, very small changes in generation time occurred for our strains, highlighting the psychrophilic and psychrotolerant nature of our organisms. At −10 °C, though, a very sharp increase in generation time occurred, but no further change occurred, even for temperatures as low as −36 °C. The sharp increase in generation time at −10 °C was seen across multiple species in multiple studies (Figure 5), with generation times in the order of a year or more for many organisms. Our experiments, performed in very low nutrient conditions, showed even longer generation values (as high as 100 years). These values are not practical for in situ growth-based detection in frozen environments, especially at the very low temperatures expected in extraterrestrial settings. This difficulty highlights the need to develop hyper-sensitive methods to detect biosignatures of active biological processes and using multiple parallel approaches. Likewise, methods for biosignature searches may need to differentiate between oligotrophic and heterotrophic environments.

### 4.4. Proteomics

In addition to basic biosignatures, such as cells (or cell fragments), activity, or even recoverable cells, proteomic analysis can provide further insight on cellular architecture, metabolic processes, and responses to environmental stress. Even though all three bacterial strains showed comparatively high metabolic activity at −5 °C, we chose Cp34H to further investigate at the molecular level due to the availability of a high-quality genome sequence and extensive investigations from the last two decades [23,37]. Our molecular approach aimed to identify newly synthesized 13C-labeled proteins in the −5 °C incubation of Cp34H, which could inform us of cellular priorities and first-response systems when supplied with minimal nutrients under cold stress.

#### 4.4.1. Motility

Flagellin was consistently abundant across all recorded time points, highlighting its essential role in *Colwellia psychrerythraea*. The use of flagella is widely recognized as a key adaptation to cold and high-pressure environments [23,67,68], facilitating motility and enabling the bacterium to navigate nutrient or chemical gradients [69], which is critical for survival in such extreme conditions.

#### 4.4.2. Storage

The next proteins to show significant synthesis with the stable isotope were two putative granule-associated proteins (Figure 4b). These proteins are thought to be involved in the formation, stabilization, or function of cellular granules, structures that store or compartmentalize important molecules such as proteins, RNA, or nutrients within the cell [67,70]. Specifically, in Cp34H, the early synthesis of putative granule-associated proteins could support prior suggestions that this is a mechanism for storing carbon and energy reserves in the form of polyhydroxyalkanoate (PHA) compounds [67]. PHA compounds are significant to Cp34H, as they are known to serve as intracellular energy sources and have been linked to pressure adaptation [71]. The Cp34H genes for granule-associated proteins (CPS4086, CPS4085, andCPS4084) are located near PHA synthesis genes, suggesting their co-regulation [67]. This highlights the organism’s strategy to adapt to extreme cold and high-pressure environments, such as those encountered in Earth’s ocean floor, by prioritizing resource storage and energy management, providing additional evidence for the co-regulation of these two metabolic processes.

In this study, the labeled synthesis of granule-associated proteins peaked at hour 4, followed by a steady increase from hour 24 to day 7, indicating a rapid turnover in response to changing energy needs as the environmental conditions change, with cells focusing on resource conservation and survival by synthesizing storage compounds like PHAs. This cyclic behavior in protein synthesis aligns with the metabolic dynamics of bacterial populations adapting to environmental stress [72].

In parallel, Acetoacetyl-CoA reductase and Acetyl-CoA acetyltransferase, enzymes involved in the leucine degradation pathway, also showed significant synthesis. These enzymes convert leucine into Acetoacetyl-CoA, a precursor for both energy production and PHA synthesis. From hour 4 to 24, their abundance steadily increased, peaking at the 168 h endpoint. This suggests that Cp34H may initially use leucine for ATP generation via the citric acid cycle, but later shifts towards PHA synthesis as a survival strategy. By synthesizing these enzymes early, Cp34H demonstrates flexibility in managing leucine either for immediate energy or for long-term storage in PHA granules, ensuring its survival in extreme environments such as cold or high-pressure conditions.

This combined evidence highlights a sophisticated mechanism by which *Colwellia psychrerythraea* not only metabolizes leucine but also efficiently stores it in PHA granules to adapt to environmental stresses.

#### 4.4.3. Future of Protein SIP in Life Detection

Mass spectrometry-based tracking of stable isotope incorporation, such as ^13^C-labeled amino acids, offers a powerful method to distinguish between extant and extinct life signals. From a method-development perspective, this is the first study based on ^13^C stable isotope labeling in bacterial growth at subzero temperatures. Two aspects were unique to these ^13^C-leucine experiments. First, it was key to be able to consistently maintain and evenly apply the target temperature through the culture. This need led us to conduct all incubations in 1.5 mL tubes, which were cooled down by inserting them in pre-cooled, steel tube racks. The second important aspect was to maintain a low-nutrient environment, while at the same time adding enough tracer that it could yield proteins detectable via mass spectrometry. By identifying active biological processes through the incorporation of isotopes into newly synthesized peptides, we can test for ongoing life, while the absence of such activity could suggest long-extinct life. This approach is complementary to the traditional and very sensitive radioisotope labeling experiments and provides additional context, as it highlights the specific metabolic pathways being utilized at the time of sampling, thereby linking environmental context or provided nutrients to the life type being monitored—a requirement in the standards of evidence for off-planet life detection [28]. This insight is crucial for understanding how organisms are responding to the unique environmental conditions, such as extreme cold or high pressure, that might exist on planets like Mars or moons like Europa.

## 5. Conclusions

Extreme niches of Earth’s cryosphere have a unique microbial ecology defined by simultaneous extremes of low temperature and high salinity. Organisms isolated from these environments can inform the search for life on icy worlds, with cellular-level responses to long-term subzero incubation, a proxy for biosignatures. We found that individual species had strain-specific responses in observable, recoverable, and metabolically active cells after a one-year incubation at various subzero temperatures (−5 °C, −10 °C and −36 °C). From these parameters, observable cells hold up as a reliable biosignature in frozen environments, especially at the lowest temperatures. Recoverability and activity varied greatly among strains, highlighting the need of testing different parameters. At −5 °C, *Psychrobacter* sp. strain 7E had the largest leucine incorporation, while Cp34H was the most active strain at −10 °C and −36 °C. Overall, H3E was the most resilient strain, as it maintained long-term recoverability across all temperatures. Cp34H and P7E, on the contrary, lost almost all recoverable cells at −5 °C and −10 °C. As such, the presence of either recoverable (culturable) or active cells, and the relationship between these two parameters, should be considered when looking for biosignatures. The very high generation times indicate that in situ growth-based detection in frozen oligotrophic environments may not be feasible with current methods, especially at the very low temperatures. This highlights the need for developing hyper-sensitive methods to detect microbial activity. The observation of newly synthesized proteins by the tracking of stable isotope incorporation (^13^C_6_-leucine) was successfully tested at low temperature, giving further insight on the relevance of motility and nutrient storage as cellular-level adaptations to low temperature. Monitoring protein synthesis with stable isotope labeling provides a promising technique to determine biosignatures for life in ice, of importance to astrobiological questions and to aid in the search for life elsewhere. Given the scarcity of detailed subzero studies or in-ice examinations in bacterial isolates, in particular those that are halophilic, this study also adds significantly to the limited body of knowledge regarding the growth and survival of marine halophilic isolates under extreme low-temperature conditions.

## Figures and Tables

**Figure 1 microorganisms-13-00251-f001:**
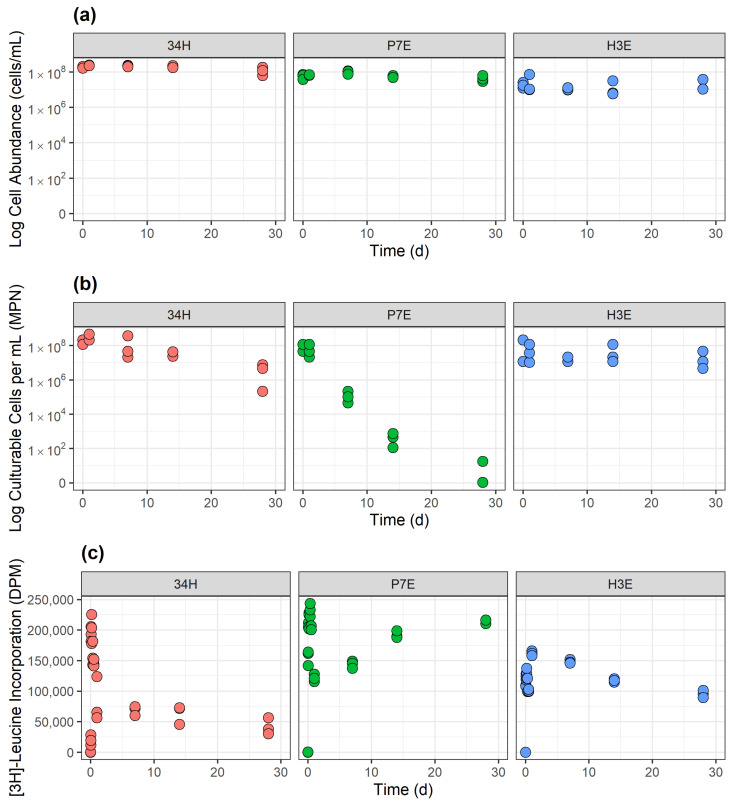
The progression of (**a**) cell abundance, (**b**) recoverability, and (**c**) metabolic activity over 1 month at control temperature for *Colwellia psychrerythraea* str. 34H (Cp34H, red circles), *Psychrobacter* sp. str. 7E (P7E, green circles), and *Halomonas* sp. str. 3E (H3E, blue circles). The control temperatures were 8 °C for (Cp34H) and 22 °C for P7E and H3E. Note that the scale in panels (**a**,**b**) is logarithmic, whereas in (**c**) it is linear. In panel (**c**), data for H3E have been adjusted as described in Section 2.3.1 to allow for the comparison of metabolic activity among strains.

**Figure 2 microorganisms-13-00251-f002:**
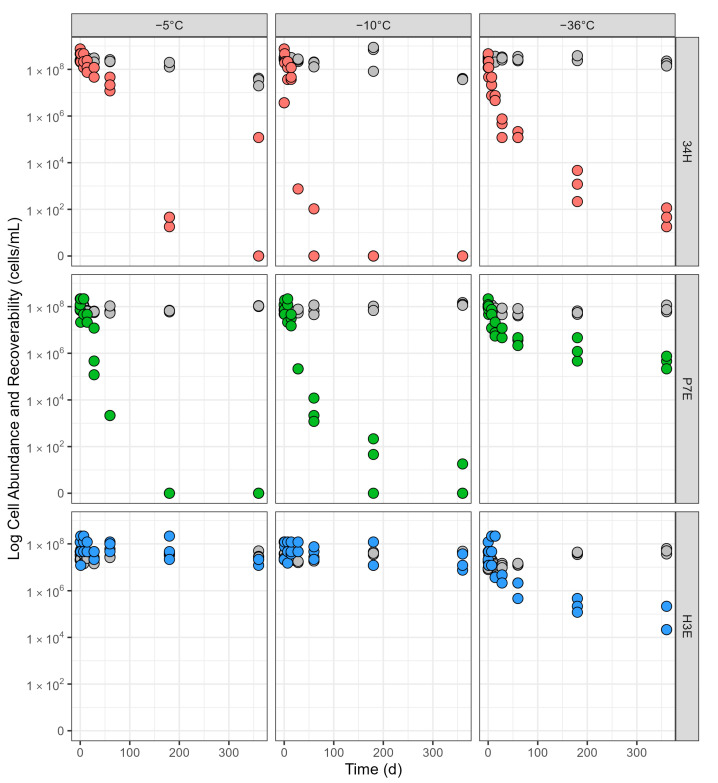
The progression of cell abundance (gray circles) and recoverability (colored circles) for all strains during the 1-year incubation at three subzero temperatures (−5 °C, −10 °C, and −36 °C). Red circles: *Colwellia psychrerythraea* str. 34H; green circles: *Psychrobacter* sp. str. 7E; blue circles: *Halomonas* sp. str. 3E. For a detailed view of the first two months, see Appendix A.

**Figure 3 microorganisms-13-00251-f003:**
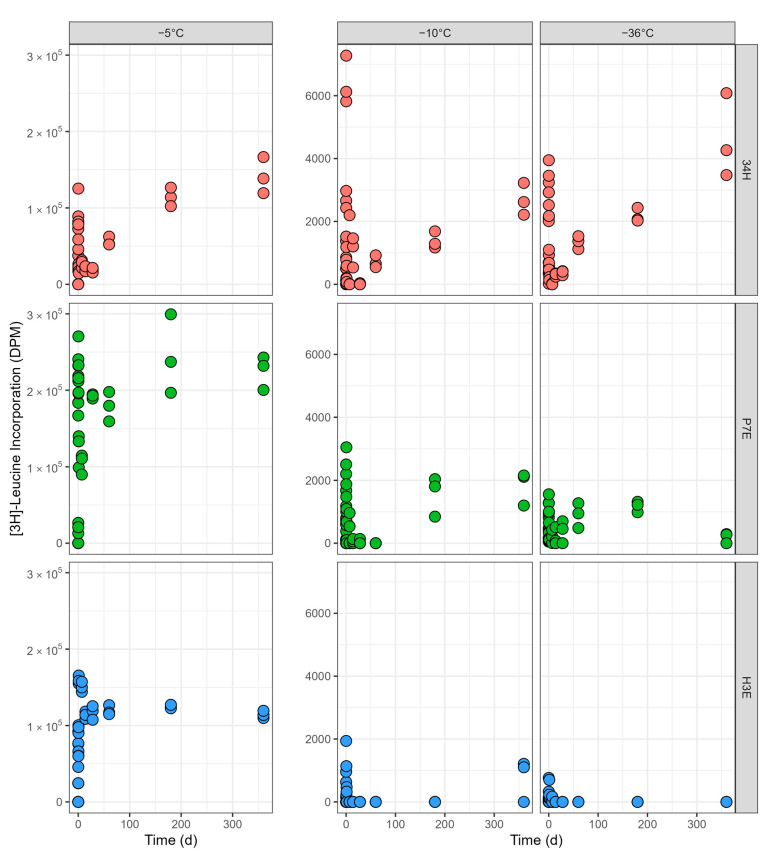
The progression of [^3^H]-leucine incorporation for all strains during the 1-year incubation at three subzero temperatures (−5 °C, −10 °C and −36 °C). Note the difference in the *y*-axis scale between −5 °C and −10 °C, −36 °C. Red circles: *Colwellia psychrerythraea* str. 34H; green circles: *Psychrobacter* sp. str. 7E; blue circles: *Halomonas* sp. str. 3E. Data for H3E have been adjusted as described in Section 2.3.1 to allow for the comparison of metabolic activity among strains. See Appendix A for a detailed view of the first day of incubation, and Appendix A for the first two months.

**Figure 4 microorganisms-13-00251-f004:**
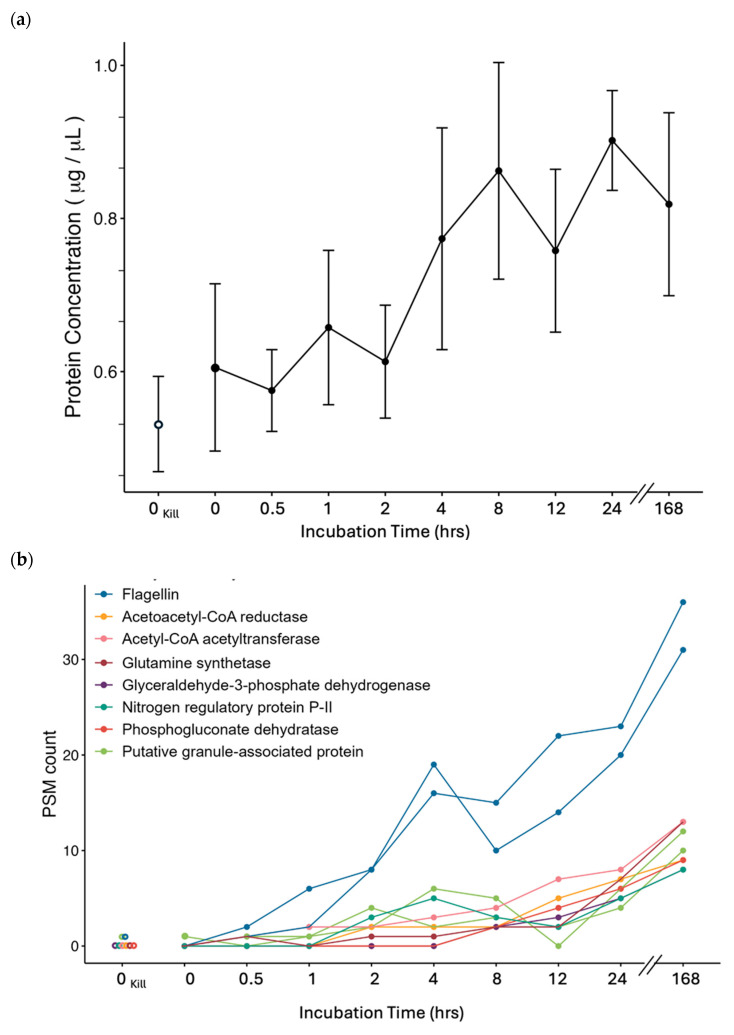
Total protein concentration (µg/µL) and the total number of spectra identified for the top 10 proteins identified, to be labeled over incubation time (hours) for the ^13^C_6_-leucine protein-SIP experiments (*n* = 5 per time point). Triplicate controls were created by adding TCA solution prior to stable isotope tracer addition (0_Kill_). (**a**) Total protein concentration was completed on all cells harvested per incubation tube, maintained at −5 °C. Protein concentrations were determined on mechanically lysed cells using a BCA protein microplate assay in µg/µL on 5 replicas per time point (averages and standard deviations are provided; killed controls (0_Kill_) represented by open circles); (**b**) protein-SIP mass spectrometry: number of peptide spectra matched (PSM count) that contain ^13^C_6_-leucine in the peptide sequence identified via proteomic tandem mass spectrometry experiments completed on each time point. PSM counts for the killed controls (0_Kill_) are represented by open circles. Two Flagellin proteins were labeled (blue: Q485N5,Q485N3), two putative granule-associated proteins (lime green: Q47WT2, Q47WT3), and one of each of the following: nitrogen regulatory protein (dk green: Q488X3), glutamine synthetase (maroon: Q489V6), Acetyl-CoA acetyltransferase (pink: Q481C9), Phosphogluconate dehydratase (red: Q482L4), Glyceraldehyde-3-phosphate dehydrogenase (purple: Q482F8), and Acetoacetyl-CoA reductase (orange: Q47WS5).

**Figure 5 microorganisms-13-00251-f005:**
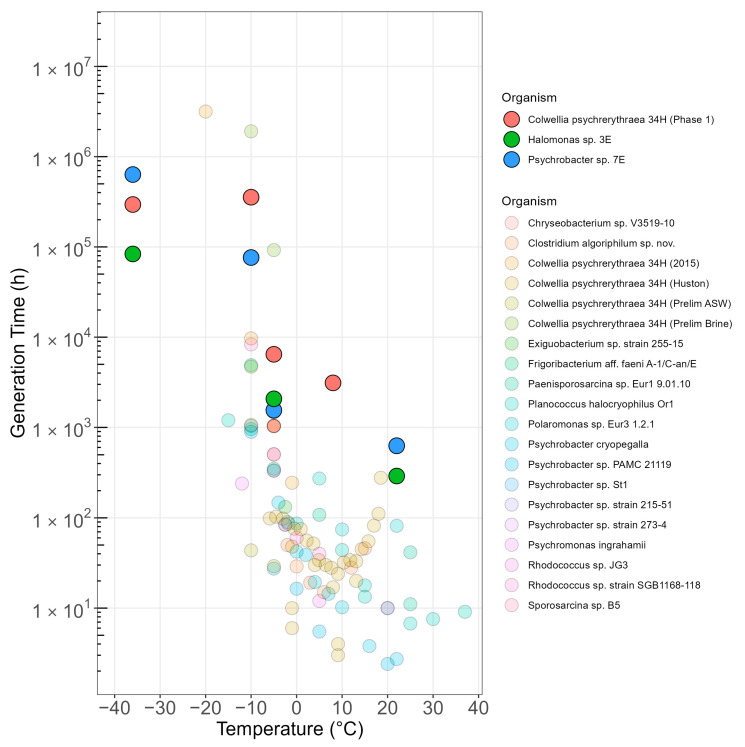
The relationship between generation time and temperature for psychrotolerant and psychrophilic cultured organisms. Strong colors indicate organisms examined in this study. Pale colors indicate organisms whose generation time was found in the literature or calculated from published values. We excluded studies utilizing environmental samples, as well as those where metabolic rates were measured but not converted to growth rate or generation time. Data provenance: Appendix B and Appendix A.

## Data Availability

The original cell abundance, recoverability, and leucine incorporation data are available at the Arctic Data Center, https://doi.org/10.18739/A2JH3D45M. The mass spectrometry proteomics data are publicly available on the ProteomeXchange Consortium via the PRIDE [73] partner repository with the dataset identifier PXD059470. Additionally, an interactive proteomic data analysis from all experiments is available on the Limelight Server (https://limelight.yeastrc.org/limelight/p/cp34h_perchlorate_13c_leucine (accessed on 6 January 2025)).

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
