# Peer review of "Metabolic Responses, Cell Recoverability, and Protein Signatures of Three Extremophiles: Sustained Life During Long-Term Subzero Incubations"

_microorganisms, 2025, doi:10.3390/microorganisms13020251_

Round 1
Reviewer 1 Report
Comments and Suggestions for Authors
The article presents valuable data on the growth of several strains at low temperatures. These data are of high interest from the point of view of understanding possibility of life existence under extremal conditions, including those on other planets. Thus, the article has high scientific significance and novelty. Methods are carefully described; they are adequate to the goals of the study. The results correspond to the conclusions. Thus, the work may be accepted after minor revisions.
Line 97. “determined in previous experiments by M. Ewert and S. Carpenter.” Could you provide reference or results of the experiment?
Line 110-123. This paragraph repeats the results of the present study. It may be excluded.
Line 274. MS data analysis – Subsection number?
Figure 4. Enlarge figure 4b and legend.
Table S1, Row “Halomonas sp. 3E -10°C”….. check the data in the row.
Author Response
Thank you for taking the time to review in detail this manuscript. We found the comments helpful and have made changes accordingly. Please find the detailed responses below and the corresponding revisions and corrections in the track-changed re-submitted files.
Comments 1: Line 97. “determined in previous experiments by M. Ewert and S. Carpenter.” Could you provide reference or results of the experiment?
Response 1: Yes, we can provide the results of the experiments and agree that having the results available can improve the manuscript. We have added the data as a supplemental table (Table S1). The numbering of other supplemental tables was modified accordingly. When preparing this data, we also noticed there was a mistake in the growth rates for Psychrobacter sp. str. P7E: its lowest measured temperature for growth is actually -8C (instead of -1C). We have corrected the text in the manuscript accordingly in Line 92.
Comments 2: Line 110-123. This paragraph repeats the results of the present study. It may be excluded.
Response 2: We agree with the reviewer. We have deleted this paragraph from the introduction, and have incorporated the quantitative information of the paragraph into the Conclusions section (with minor amendments of the conclusions to improve readability).
Comment 3: Line 274. MS data analysis – Subsection number?
Response 3: Thank you for pointing this out. Subsection number added.
Comment 4: Figure 4. Enlarge figure 4b and legend.
Response 4: We agree with the reviewer and have changed the layout of figure 4 in order to enlarge the figure and legend. The original figure has been replaced. Please note that this change was done without the automatic track change. We have also fixed minor typos in the caption.
Comment 5: Table S1, Row “Halomonas sp. 3E -10°C”….. check the data in the row.
Response 5: Thank you for pointing this out. Row deleted. Please note the values for Halobacter sp. strain H3E have been updated in the supplementary table as a response to another reviewer’s observation. We also made changes to make the decimal format homogeneous (no decimals on values of generation time above 100 h)
Reviewer 2 Report
Comments and Suggestions for Authors
In a publication entitled 'Metabolic Responses, Cell Recoverability, and Protein Signatures of Three Extremophiles: Sustained Life During Long Term Sub-Zero Incubations’, described a very interesting perspective on research into the effects of low temperatures on the growth and protein metabolism of micro-organisms isolated from extreme environments. The most interesting aspect seems to be the context in which the work was presented. That is, the potential exploration of the closest star system to us and the extreme conditions found there.
I present a review of the work below:
The work presented for review is on the borderline between experimental, review and theoretical work. However, this does not exclude it from publication in the submitted journal. The results contained in the paper, combined with an analysis of the results already achieved by the authors and an indication of potential problems that may be encountered in the future, the authors have produced a paper worthy of attention.
Introduction.
In the introduction, the authors have outlined a very interesting theory of the conditions that exist in space. However, in the context of this research, it is important to also outline the potential benefits that protein metabolism research will provide in the context of the theory presented. The authors should add some information on this topic in the context of the known literature.
Lines 98-109 - This section is more appropriate to the methodology than the introduction.
Lines 110-123 - This passage fits better in the summary.
At the end of the introduction, it would be useful to state the aim/hypothesis of the study.
Methodology
Chapter 2.1.2 - No subscripts in substrate components
Chapter 2.1.4 - How did the authors determine the optimal growth temperatures of the isolates studied? Were they derived from previous studies?
Due to the wide variation in sampling dates for the study and the variation in the different stages of the study, the authors could have included an additional timeline indicating when and which samples were taken for which experiments.
Section 2.3.1 - Why did the authors use different concentrations and isotopes of leucine for different strains? The leucine isotope was only added at time T0 and observed over time?
Section 2.4 - Why was the Cp34H strain chosen for this experiment?
The results
Due to the very wide and varied time range of the samples taken, the graphs from the early stages of the study overlap heavily, affecting the appearance and readability of the graphs. Are the authors able to improve the readability of the graphs presented?
From what the authors describe in the results, some strains, e.g. P7E, maintain a similar cell count in the DAPI test, but after 6 months they lose recoverability but still incorporate the leucine isotope into their proteins? Is there any explanation for this situation?
Author Response
Thank you for taking the time to review this manuscript and contributing to the text, data interpretation, editorial details and helping us improve the big-picture presentation of our results. We found the comments useful and have addressed them as detailed below. Please find the corresponding revisions in the resubmitted track-changed files.
Comments 1: In the introduction, the authors have outlined a very interesting theory of the conditions that exist in space. However, in the context of this research, it is important to also outline the potential benefits that protein metabolism research will provide in the context of the theory presented. The authors should add some information on this topic in the context of the known literature.
Response 1: We agree with the reviewer that the introduction lacks information on how metabolic activity can be used as a biosignature, and have added additional information to the introduction.
Comments 2: Lines 98-109 - This section is more appropriate to the methodology than the introduction.
Response 2: Thank you for the suggestion. We partially agree. While we agree that the paragraph has details that belong to the methods, we think some of the information should stay in the introduction, in particular with regards to comment 5 (aims of the project). The specific temperatures were chosen to represent three environments with distinct physical characteristics. We believe it is important to have this information upfront in the introduction as it is part of our main objectives and rationale for the experiment. We have reviewed the paragraph to focus on the reasons for the choice of temperatures, minimizing the amount of information that is repeated in the methods.
Comment 3: Lines 110-123 - This passage fits better in the summary.
Response 3: Thank you for the suggestion. We have deleted this paragraph from the introduction, and have incorporated the pertinent information into the Conclusions section (with minor amendments of the conclusions to improve readability).
Comment 4: At the end of the introduction, it would be useful to state the aim/hypothesis of the study.
Response 4: We agree with the reviewer. We have stated the aim and hypothesis of the project.
Comment 5: Chapter 2.1.2 - No subscripts in substrate components
Response 5: Thank you for bringing this up. We have fixed the missing subscripts.
Comment 6: Chapter 2.1.4 - How did the authors determine the optimal growth temperatures of the isolates studied? Were they derived from previous studies?
Response 6: The optimal temperature for Cp34H is well known from previous studies. We have added a reference in section 2.1.4. For P7E and H3E, we have laboratory data with initial limits of temperature and salinity growth, assessed by turbidity. For the control treatment, we chose the fastest, known growth temperature. We have included the results from those experiments as part of the supplementary material (Table S1), as requested by another reviewer. We have also replaced “optimal temperature” by “control temperature” for P7E and H3E in the text.
Comment 7: Due to the wide variation in sampling dates for the study and the variation in the different stages of the study, the authors could have included an additional timeline indicating when and which samples were taken for which experiments.
Response 7: The experiment was divided in phases, where the incubations for each strain were setup with 2 months difference (34H experiments were set up on March, P7E on May and H3E on July 2019). A full calendar with the dates of collection, and the specific samples collected each day, has been made available as Chart S1.
Comment 8: Section 2.3.1 - Why did the authors use different concentrations and isotopes of leucine for different strains? The leucine isotope was only added at time T0 and observed over time?
Response 8:
The leucine isotope was only added at time T0 and its incorporation in cells was observed through time by sampling 3 replicas at each data point (all replicas for all time points were setup the same day).
As for the first question, we thank the reviewer for bringing this up to our attention. The use of different 3H leucine reagents for the activity measurements was due to constrains regarding the stocks available in the laboratory at the time of the experimental setup. We reviewed our procedures and found an inconsistency in our data, described and corrected as follows:
During the experiment, all working stocks were normalized to microcuries to have similar conditions regarding work with radioactive substances. However, normalization to microcuries implied that the strain that received the 149 Ci mmol-1 stock (H3E) received a lower concentration of leucine. For a direct comparison among strains, H3E data had to be divided by 2.754 (the ratio between the radioactivity of the two tracers). We have corrected Fig 1c, Fig 3, Fig 6 and the corresponding Supplemental Table (Table S2 in the resubmission).
To reflect this on the text, we have added a note on the methods section (section 2.3.1), the captions of the figures, and everywhere that leucine incorporation data for H3E was mentioned. The only major change is that H3E is no longer the more active strain, it is P7E.
Comment 9: Section 2.4 - Why was the Cp34H strain chosen for this experiment?
Response 9: We focused on Cp34H as it is the only one with an available, high-quality, genome sequence, which would allow the proteomic analysis. The other strains do not have a high-quality genome available. We have added this information to the first paragraph of section 2.4.
Comment 10: Due to the very wide and varied time range of the samples taken, the graphs from the early stages of the study overlap heavily, affecting the appearance and readability of the graphs. Are the authors able to improve the readability of the graphs presented?
Response 10: We agree with the reviewer on the problem of overlapping data. During the preparation of the manuscript we tried multiple variations of the figures, including having different figures for different time ranges. We settled on the current presentation as a compromise in the number of figures and the relevance of the data to support our analysis and conclusions. As a solution, we have included supplementary figures with additional time ranges: Figure S1 will present cell number and recoverability data for the first two months. Figure S2 will present activity data for the first day, and Figure S3, the activity data for the first two months. A reference to the supplementary figures has been included in sections 3.1.2, 3.1.3 and 3.1.4
Comment 11: From what the authors describe in the results, some strains, e.g. P7E, maintain a similar cell count in the DAPI test, but after 6 months they lose recoverability but still incorporate the leucine isotope into their proteins? Is there any explanation for this situation?
Response 11: This is an insightful question. Microorganisms in our experiments have levels of leucine incorporation associated with maintenance metabolism (Price & Sowers 2004; maintaining homeostasis, turnover of macromolecules, repair enzymes). Our data seem to show that more metabolically active cells (such as P7E) had more difficulty being recovered; one explanation is that their active metabolism may have led to more accumulation of metabolic byproducts, and as such, the need of additional recover/detoxification time and perhaps different (low nutrient) growth conditions. This interpretation is based on literature results where cells frozen for an extended period of time seem to need additional time and low-media culture to repair and detoxify before they can resume reproduction (e.g. Christner et al, 2000). Our experiments did not test if cells would improve their recoverability after allowing for an extended repair period, but the relationship between activity and recoverability is intriguing, and possibly a future avenue of research.
In response to the reviewer’s question, we have added at the end of section 4.1 additional information on how an increased metabolic activity may difficulty of recovery of cells and the implications this can have for recovery efforts of potential extraterrestrial life.